# Signaling Pathways in Human Blastocyst Development: From Molecular Mechanisms to In Vitro Optimization

**DOI:** 10.3390/jdb13030033

**Published:** 2025-09-09

**Authors:** Yan Jiao, Jiapeng Liu, Congge Li, Yuexin Hu, Sanjun Zhao

**Affiliations:** 1School of Life Sciences, Yunnan Normal University, Kunming 650500, China; 18178908316@163.com; 2College of Life Sciences, Southwest United Graduate School, Kunming 650092, China; 3Medical School, Kunming University of Science and Technology, Kunming 650500, China; ljp200009@163.com; 4State Key Laboratory of Primate Biomedical Research, Institute of Primate Translational Medicine, Kunming University of Science and Technology, Kunming 650500, China; lcg17838905082@163.com; 5Yunnan Key Laboratory of Primate Biomedical Research, Kunming 650500, China; 6School of Humanities and Management, Kunming Medical University, Kunming 650500, China; 7Yunnan International Joint R&D Center for Sustainable Development and Utilization of Biological Resources, Kunming 650500, China

**Keywords:** assisted reproductive technology, preimplantation embryo development, signaling pathways, lineage specification

## Abstract

In recent years, assisted reproductive technology (ART) has developed rapidly with the delay in reproductive age and the rise in infertility rates. During ART, blastocyst quality is a key factor affecting the rate of implantation and clinical pregnancy, and blastocyst formation is dependent on the precise regulation of multiple signaling pathways in preimplantation embryo development. In this review, we systematically analyze the molecular mechanisms of the core pathways, including Hippo, Wnt/β-catenin, FGF, Nodal, and BMP, in blastocyst lineage differentiation and morphogenesis, and assess the feasibility of optimizing in vitro culture by targeting key signaling nodes, as well as provide theoretical support for constructing research models of preimplantation embryos.

## 1. Introduction

Infertility is a highly prevalent global health condition. It is estimated that between 8% and 12% of couples of reproductive age worldwide suffer from infertility [1]. Its prevalence exhibits marked regional variation, with infectious tubal factors being the predominant cause in low-income countries [2], while delayed childbearing, environmental exposures, and metabolic disorders are more common contributors in high-income countries. This epidemiological landscape has given rise to the rapid development of assisted reproductive technology (ART), a suite of techniques involving the in vitro manipulation of human gametes or embryos to achieve pregnancy, including in vitro fertilization (IVF), intracytoplasmic sperm injection (ICSI), and preimplantation genetic testing (PGT) [3]. Since the birth of the first IVF-conceived individual in 1978, more than 8 million ART offspring have been born worldwide [4,5]. However, ART clinical pregnancy rates are still limited by embryo quality. Current data indicate that only approximately 50% of embryos cultured in vitro progress to the blastocyst stage suitable for transfer. Among these, high-quality blastocysts achieve implantation rates of up to 72.8%, whereas low-quality counterparts show rates as low as 28.1%. [6,7,8]. Therefore, optimizing the in vitro culture system to enhance the quality of blastocysts has become the core direction to break the bottleneck of ART efficiency. The realization of this goal depends on the in-depth analysis of the regulatory mechanisms of embryo development before implantation.

Preimplantation embryonic development is a highly programmed biological process that involves key events such as zygotic genome activation (ZGA), the establishment of cell polarity, and lineage specification, ultimately resulting in the formation of a blastocyst composed of the epiblast (EPI), trophectoderm (TE), and primitive endoderm (PrE) [9,10]. This dynamic process not only determines the developmental potential of the embryo but also directly influences subsequent implantation, pregnancy outcomes, and even the health of the offspring [11,12,13]. Increasing evidence indicates that proper embryonic development during this stage relies on the precise coordination of multiple signaling pathways, such as Hippo, Wnt, TGF-β, etc. [14]. Meanwhile, disruptions in these regulatory networks are closely associated with developmental arrest or morphological abnormalities [15,16,17]. Thus, targeting critical nodes within these pathways may provide new strategies to improve embryo quality.

In this review, we systematically summarize and analyze the regulatory roles of multiple signaling pathways in preimplantation embryonic development, with a focus on their contributions to blastocyst formation. On this basis, we evaluate the potential value of small-molecule compounds’ targeted intervention strategies in optimizing blastocyst development, and further explore how these pathways orchestrate lineage specification at the molecular level, thereby providing a theoretical foundation for improving in vitro culture systems and enhancing clinical outcomes in ART.

## 2. Preimplantation Human Embryonic Development

Human preimplantation embryonic development is a highly orchestrated process lasting seven days, from fertilization to the point at which the blastocyst implants into the endometrium (Figure 1) [18,19]. This stage is marked by critical events—including cell proliferation, morphological remodeling, and lineage specification—that collectively determine the embryo’s developmental competence and implantation potential. During preimplantation development, the embryo undergoes several rounds of symmetric cleavage to form the two-cell, four-cell, and eight-cell stage embryos. At the eight-cell stage, ZGA drives the transition from maternal transcript dependence to embryo-specific gene expression in the embryo [20]. Around d.p.f 3 to 4, the embryo enters the morula stage, during which enhanced cell–cell adhesion mediated by E-cadherin leads to compaction [21]. Concurrently, the establishment of apical–basal polarity drives the first cell fate decisions, enabling blastocoel formation via cavitation [22,23]. By d.p.f 6-7, the blastocyst hatches from the zona pellucida and gains the capacity to initiate implantation through interaction with the maternal endometrium [24,25].

During this process, the embryo undergoes a total of two successive lineage segregations. The first lineage specification coincides with blastocoel formation, where outer polar cells differentiate into the TE, which contributes to placental structures, while inner apolar cells form the inner cell mass (ICM), which gives rise to the fetus and portions of the extraembryonic tissue. As the blastocyst matures, the ICM undergoes a second lineage segregation: cells adjacent to the polar TE differentiate into the EPI, the progenitor of the embryo proper and amniotic tissue, while the remaining ICM cells form the PrE, which develops into the yolk sac [26,27,28].

## 3. Signal Pathway in Human Preimplantation Embryo Development

We systematically reviewed the regulatory roles of the Hippo, Wnt, FGF, and TGF-β signaling pathways during human preimplantation embryo development. Furthermore, to assess the feasibility of improving in vitro embryo outcomes through targeted modulation of these pathways, we further summarized the application of various signaling regulators in human preimplantation embryo studies (Table 1).

### 3.1. Hippo Signaling Pathway: Regulating TE Differentiation

The Hippo pathway is a highly conserved signaling cascade centered on a serine/threonine kinase core that negatively regulates the expression and activity of the transcriptional coactivators YAP and TAZ. In mammals, key components of this pathway include MST1/2, SAV1, MOBKL1A/B, LATS1/2, YAP/TAZ, and their downstream transcription factors TEAD1–4 [37,38]. When the pathway is active, YAP and TAZ are phosphorylated and retained in the cytoplasm, thereby losing their transcriptional regulatory function. In contrast, when the pathway is inhibited, dephosphorylated YAP/TAZ translocate into the nucleus, where they interact with TEAD transcription factors to activate the expression of target genes. Through this mechanism, the Hippo pathway participates in regulating organ size, tissue regeneration, and tumorigenesis [39,40].

During preimplantation embryonic development in mammals, the Hippo signaling pathway is widely recognized as a key determinant of the fate decision between the ICM and the trophectoderm TE [41]. In mice, at the eight-cell stage, as cell polarity is established, apical polarity complexes (such as aPKC) in the outer polarized cells sequester LATS1/2 and angiomotin (AMOT) to the apical domain, rendering them inactive. As a result, the Hippo pathway is suppressed, allowing YAP/TAZ to translocate into the nucleus, where they interact with TEAD4 to activate TE-specific genes such as CDX2 and GATA3, thereby driving TE lineage specification. In contrast, in the inner non-polarized cells, AMOT activates LATS1/2 through phosphorylation, leading to Hippo pathway activation. YAP/TAZ are phosphorylated and retained in the cytoplasm, suppressing TE-related gene expression and promoting the expression of ICM markers such as NANOG and SOX2, thus facilitating ICM lineage commitment [42,43,44,45,46].

In human embryos, the role of the Hippo signaling pathway in the first lineage specification is partially conserved but also exhibits notable species-specific differences. Studies have shown that during the morula stage, the apical membranes of outer cells in both human and mouse embryos display similar localization patterns of aPKC and AMOT, suggesting that the Hippo pathway is suppressed in the outer cells of both species. Moreover, the inhibition of aPKC in human embryos leads to reduced nuclear YAP levels in outer cells, further supporting the role of this pathway in TE formation [29]. However, the involvement of the Hippo pathway in ICM specification differs between humans and mice. In early human blastocysts, SOX2 is expressed in both inner and outer cells, whereas in mice it is restricted to inner cells. This indicates that the YAP/TEAD axis directly regulates SOX2 expression in mice, while its regulatory role in human embryos may be altered [29]. Additionally, while TEAD4 knockout in mice leads to the downregulation of CDX2 and GATA3, causing developmental arrest at the morula stage and preventing blastocyst formation, in human embryos, TEAD4 knockout similarly reduces CDX2 expression but does not affect GATA3, and blastocoel formation still proceeds [47,48,49]. These findings suggest that ICM specification in human embryos may involve downstream effectors or compensatory mechanisms beyond the TEAD4/CDX2 axis.

The above findings highlight the critical role of the Hippo signaling pathway in the first lineage segregation of human embryos; however, its influence is not limited to this developmental stage. For instance, during preimplantation development, the core Hippo components YAP and TEAD4 may contribute to zygotic clearance of maternal mRNAs. Impaired zygotic degradation has been associated with developmental arrest at the eight-cell stage in human embryos [50]. Furthermore, at the blastocyst stage, TEAD1 and YAP1 are co-localized in both trophectoderm and primitive endoderm precursor cells, suggesting that the Hippo pathway may also be involved in the second lineage segregation [51].

In studies investigating the Hippo signaling pathway during human preimplantation embryo development, Gerri et al. activated Hippo signaling by adding the aPKC inhibitor CRT0276121 from the late eight-cell stage to the morula or blastocyst stage. This activation led to the reduced nuclear localization of YAP1 and downregulation of GATA3 expression in outer cells. Moreover, most embryos treated with the inhibitor arrested at the morula stage and failed to undergo cavitation [29]. Conversely, targeted inhibition of LATS kinases (large tumor suppressor kinases) during the same developmental window blocked the Hippo pathway and promoted TE differentiation. Although SOX2 expression was partially downregulated, blastocyst formation was not affected [30]. These findings suggest that in embryos with poor developmental potential or difficulty forming a blastocoel, moderate inhibition of key Hippo signaling components may facilitate blastocyst formation.

### 3.2. Wnt Signaling Pathway: Disruption Impairs Trophectoderm Formation

The Wnt signaling pathway is a highly conserved regulatory system that plays a critical role in embryonic development. It is mainly composed of the canonical Wnt/β-catenin pathway and non-canonical Wnt pathways. In the canonical Wnt/β-catenin pathway, β-catenin functions as the key effector molecule, and its modification and degradation are pivotal events in pathway regulation. In the absence of Wnt ligands, the Axin/GSK3/APC complex promotes the degradation of β-catenin. When Wnt is present, the formation of the Axin/GSK3/APC complex is blocked, thereby preventing β-catenin degradation. As a result, β-catenin accumulates in the cytoplasm and translocates into the nucleus, where it interacts with TCF/LEF1 transcription factors to initiate the expression of target genes [52,53,54].

Several studies have demonstrated the role of Wnt signaling in the establishment of the body axis during mammalian embryogenesis [55,56]; however, its specific regulatory mechanisms during the blastocyst formation stage remain poorly characterized. In mouse embryos derived from IVF, the inhibition of the Wnt pathway has been reported to promote the formation of the PrE, thereby promoting embryonic development [57]. A similar phenomenon has been observed in goat embryos, where Wnt pathway inhibition at the morula stage significantly increases the blastocyst formation rate [58]. In contrast, the sustained activation of Wnt signaling markedly disrupts embryonic development and impairs normal hatching in both mouse and bovine embryos [59,60,61]. However, studies in primates show a very different pattern of regulation. For example, the inhibition of Wnt signaling in marmoset preimplantation embryos interferes with the segregation of NANOG and GATA6, blocks PrE fate determination, and inhibits further embryo development, indicating a regulatory mechanism opposite to that observed in mice and goats [15]. Single-cell transcriptomic analyses have shown that multiple Wnt pathway agonists are upregulated in the human EPI, consistent with patterns observed in marmosets, suggesting that the role of Wnt signaling in human preimplantation development may more closely resemble that of primates rather than rodents [62]. The current evidence indicates that Wnt signaling promotes the exit from pluripotency in human embryos during the peri-implantation period [63]. However, its precise role in regulating EPI development during the blastocyst stage remains to be elucidated and warrants further investigation.

Wnt signaling ligands are broadly expressed across all lineages during human preimplantation embryo development [64]. At this stage, β-catenin is primarily localized in the submembranous cytoplasmic region of embryonic cells [31]. Studies have shown that the activation of Wnt signaling by the exogenous Wnt3 protein or a GSK3β inhibitor can promote the formation of trophectoderm precursor cells (TBs). However, both the activation and inhibition of Wnt signaling lead to the suppression of TE cell fate [31]. These findings highlight the critical role of the Wnt/β-catenin pathway in regulating the first cell fate decision during human embryonic development, although the underlying mechanisms responsible for this phenomenon remain to be fully elucidated.

### 3.3. FGF Signaling Pathway: A Central Driver of Hypoblast Formation

Fibroblast growth factors (FGFs) and their tyrosine kinase receptors (FGFRs) play critical roles in various developmental processes in mammals, including the proliferation, survival, migration, and differentiation of embryonic stem cells [65]. Upon ligand binding, FGFRs undergo dimerization and autophosphorylation on intracellular tyrosine residues, thereby initiating downstream signaling cascades. FGF signaling exerts its biological functions primarily through four classical pathways: the Ras/Raf-MEK-mitogen-activated protein kinase (MAPK) pathway, the phosphatidylinositol 3-kinase/protein kinase B (PI3K/AKT) pathway, the phospholipase Cγ (PLCγ) pathway, and the signal transducer and activator of transcription (STAT) pathway [66].

Existing studies have demonstrated that the FGF signaling pathway plays an important role in the second lineage differentiation of mammalian embryos [66,67,68,69]. Following the completion of the first lineage segregation, ICM cells differentiate into different cells at different internalization times. Early internalized cells express higher levels of the FGF4 ligand, while later internalized cells exhibit elevated levels of FGF receptor 1 (FGFR1) and FGFR2. The binding of the FGF4 ligand to FGFR1 and FGFR2 on cells activates downstream FGF signaling, leading to differential expression of the genes *Gata6* and *Nanog* among the cells. Cells expressing *Nanog* develop into EPI whereas cells expressing the *Gata6* gene eventually differentiate into the PrE [70,71,72,73]. This process is the second lineage specification. Therefore, the ratio of EPI to PrE cells can be modulated by adjusting the concentration of FGF4 [74]. The exogenous addition of FGF4 promotes the differentiation of ICM cells toward the PrE lineage, whereas the inhibition of FGF signaling facilitates ICM cells’ conversion into EPI [75,76]. After PrE/EPI segregation is completed, FGF signaling activity begins to decline [77].

However, the dependence of PrE fate specification on FGF signaling is not as prominent in other mammals such as pigs, bovines, and rabbits. Unlike in mice, where blocking FGF/ERK signaling at the eight-cell stage leads to the elimination of GATA4/6 expression, the inhibition of FGF signaling in early embryos of non-rodent species does not directly disrupt PrE lineage specification [78,79,80].

Current evidence suggests that FGF signaling plays a stage-specific and dose-dependent role in the specification of the PrE lineage in human embryos. In an early study, Roode et al. treated human embryos from d.p.f 3 to d.p.f 6/7 with 1.0 μM PD0325901, 0.1 μM PD173074, or a combination of both to inhibit the FGF/ERK signaling pathway. Despite this treatment, GATA4^+^ and SOX17^+^ PrE cells were still abundantly present and mutually exclusive with NANOG^+^ EPI cells, suggesting that PrE formation under these conditions may occur independently of FGF signaling [32]. However, it is worth noting that the concentration of the FGFR inhibitor used in that study (0.1 μM) was below the effective threshold (0.5 μM) required to suppress PrE formation. This limited effect may be due to a signaling barrier posed by the structural encapsulation of ICM by the surrounding TE, which could hinder the accessibility of inhibitors to the ICM. Consequently, higher inhibitor concentrations may be necessary to effectively perturb PrE fate in human embryos. Supporting this notion, Dattani et al. treated embryos at the onset of d.p.f 5 blastocyst expansion with 0.5 μM PD173074 for 24–44 h, which significantly suppressed the expression of PrE markers, and the proportion of SOX17^+^/FOXA2^+^ double-positive cells decreased from 14.2% to 1.7%, indicating a crucial role for FGF signaling in PrE induction. Conversely, the addition of 250 ng/mL FGF2 increased the proportion of PrE cells from 15.6% to 33.2%, accompanied by a reduction in EPI cell numbers [33]. In summary, although early studies did not observe a clear effect of FGF signaling inhibition on PrE formation, more precise timing and optimized drug concentrations have shown that both the activation and inhibition of the FGF pathway significantly influence PrE lineage establishment. Therefore, systematic studies exploring various concentrations and intervention windows are still needed to fully understand the role of FGF signaling in regulating PrE fate.

### 3.4. TGF-β Signaling Pathway: Maintaining EPI Stability and Regulating Apoptosis

The transforming growth factor β (TGF-β) signaling pathway family consists of two key branches, the TGF-β/Activin/Nodal pathway and the bone morphogenetic protein (BMP) pathway. These pathways function through the activation of distinct SMAD cascades, with TGF-β/Activin/Nodal signaling activating SMAD2/3 and BMP signaling activating SMAD1/5/9 [81,82,83]. The study demonstrated that these two branches have important regulatory functions during preimplantation embryonic development.

The Nodal pathway shows significant activity at the human blastocyst stage by activating Smad2/3-mediated transcriptional regulation through binding to type I and type II serine/threonine kinase receptors [84,85]. Although Nodal signaling components can be detected in mouse preimplantation embryos, studies of their function have mostly focused on the post implantation stage [86,87]. Species differences are even more pronounced in stem cell models: mouse embryonic stem cells (mESCs) do not require Nodal signaling to maintain pluripotency, whereas porcine induced pluripotent stem cells (iPSCs) and human embryonic stem cells (hESCs) rely on this pathway to activate core pluripotency factors such as NANOG, suggesting that Nodal signaling has a species-specific role in regulating the pluripotency network in humans [88,89,90,91].

Current studies suggest that, under specific conditions, the inhibition of Nodal signaling may promote the differentiation of the human EPI lineage. In a 2014 study, Van et al. treated d.p.f 3 to d.p.f 6 human embryos with 10 μM of the TGF-β inhibitor SB431542 for 72 h and observed a significant increase in the number of NANOG-positive cells within the ICM, indicating that the suppression of this pathway may facilitate EPI expansion [34]. More recently, a 2025 study by Brumm et al. employed a short-term treatment with a higher concentration of the Nodal pathway inhibitor A8301, which markedly reduced Nodal activity, but no change in the number of EPI cells was observed. However, NANOG expression was upregulated under this treatment condition, consistent with the findings of Wang et al. Therefore, we propose that the strength of TGF-β signaling is closely associated with the stability of EPI pluripotency [35,64]. In addition, the embryos used by Van et al. were low-quality D2–D3 embryos (with ≥35% fragmentation and not meeting standard cryopreservation criteria) [34]. Under these conditions, inhibitor treatment showed a tendency toward phenotypic improvement. This study provides a potential direction for optimizing the in vitro culture of poor-quality embryos and suggests that modulating Nodal signaling may support their EPI development.

As the other branch of the TGF-β superfamily, the BMP signaling pathway plays a critical role in dorsal–ventral axis formation and cell fate determination during vertebrate embryonic development [92,93,94]. In mammals, BMP signaling is primarily involved in regulating the lineage differentiation of the TE and PrE. Studies have shown that BMP signaling is active from the morula to blastocyst stages in mouse embryos, with stronger activity observed in the ICM. The inhibition of BMP signaling at the eight-cell stage of mouse embryos significantly slows cell division, reduces the total cell number in the blastocyst, and prevents more than 50% of embryos from reaching the blastocyst stage. In contrast, inhibition at the late morula or blastocyst stage has a much smaller effect. Furthermore, knockdown of Smad4 or Bmpr2 leads to a marked reduction in the number of TE and PrE cells, while having no apparent impact on EPI development [95,96].

Although BMP signaling does not appear to directly alter lineage specification trajectories in human embryos, it may affect blastocyst stability and formation efficiency by modulating the mitochondria-mediated apoptotic pathway. In one study, treatment with 100 ng/mL BMP4 from d.p.f 3 for 72 h resulted in blastocyst formation in only 17.4% of embryos, significantly lower than the 61.5% observed in the control group. Moreover, most blastocysts in the treatment group exhibited structural collapse and degeneration at later stages. Despite no significant changes in the expression of lineage markers such as NANOG, CDX2, GATA3, and GATA6, BMP4 treatment markedly induced apoptosis [36]. Mechanistic investigation revealed that BMP4 downregulated the expression of SIRT1, a deacetylase of P53, while increasing the acetylation level of P53, which was predominantly localized to the mitochondria [36,97]. These findings suggest that BMP4 may impair blastocyst development by activating a mitochondrial-dependent apoptotic pathway. Further elucidation of the dual roles of BMP signaling in cell differentiation and apoptosis could provide mechanistic insights into preimplantation developmental failure in human embryos and help identify potential predictive biomarkers.

### 3.5. Signaling Networks in Early Human Embryo Development

In addition to the Hippo, the Wnt/β-catenin, FGF, Nodal, and BMP pathways mentioned above, several other signaling pathways play important roles in human preimplantation embryonic development. For example, the supplementation of IGF-1 in vitro enhances human blastocyst formation efficiency, suppresses apoptosis, and promotes ICM expansion, likely through mTOR pathway activation [98,99]. It has also been shown that although the regulatory roles of some signaling pathways are not clear, they have been explored in stem cell-related studies. For example, the AMPK signaling pathway can regulate downstream signaling pathways by sensing pyruvate induction, which drives hESC differentiation toward mesoderm [100]. In addition, the inhibition of the PI3K/AKT signaling pathway in mouse embryos induces programmed cell death and leads to a significant delay in blastocyst hatching [101,102]. The mechanisms of the role of these signaling pathways in the preimplantation development of human embryos have not been investigated in detail.

Moreover, these signaling pathways do not operate in isolation, but form a dynamically balanced signaling network through multilevel and multinodal interactions to ensure the normal development of the embryo from the fertilized egg to the blastocyst stage (Figure 2). For example, the TGF-β and FGF signaling pathways synergistically maintain EPI pluripotency and promote TE maturation in the early development of human aneuploid embryos [64]. Meanwhile, experiments on mice and hESCs have revealed that the Wnt/Nodal/BMP signaling cascade found in mouse epiblasts is conserved in hESCs, where the BMP4 pathway activates Wnt signaling, which in turn activates the Nodal pathway [103,104,105]. However, most of the existing studies on the interaction of the signaling pathways in the embryo have been carried out on mouse embryos or hESC models, so it remains unclear whether these pathways interact in the same manner during actual human preimplantation embryonic development. Nevertheless, it is undeniable that this process is highly complex; the timing, amplitude, and simultaneous presence of different signaling pathways directly influence the ultimate fate decisions of cells within the embryo, as evidenced by experiments in hESCs [105]. Therefore, more comprehensive and refined experiments are needed to explore these interactive mechanisms, which are critical for advancing our understanding of early human embryonic development.

## 4. Conclusions

In this review, we systematically analyze the critical roles of multiple signaling pathways, including Hippo, Wnt/β-catenin, FGF, Nodal, and BMP, in human preimplantation embryo development, highlighting their species-specific regulatory mechanisms in lineage specification and morphogenesis. By integrating small-molecule intervention studies, we demonstrated the potential of targeting regulatory pathways to optimize blastocyst quality in vitro, and highlighted the decisive role of the type of signaling pathway modulators, their concentration gradients, and the timing of their application in determining embryonic developmental outcomes. These insights not only improve our understanding of early human embryogenesis but also provide a theoretical basis for improving ART outcomes and developing physiologically relevant in vitro models.

## Figures and Tables

**Figure 1 jdb-13-00033-f001:**
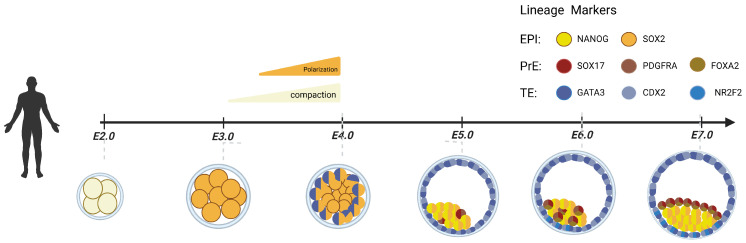
Schematic of human preimplantation development. Illustrating the timing of each morphological stage, the temporal expression of known lineage markers, and a proposed model of human lineage specification. The axis indicates the embryonic day (E) of the associated event. (Created in Biorender.com by Yan Jiao).

**Figure 2 jdb-13-00033-f002:**
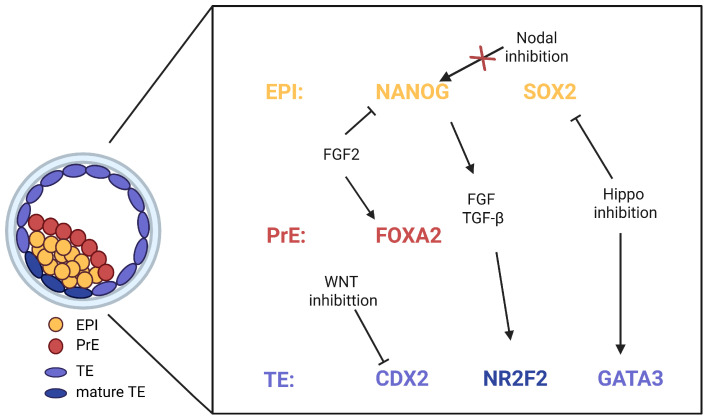
Overview of the roles of Hippo, Wnt/β-catenin, FGF, and TGF-β signaling pathways in regulating human preimplantation embryo development. EPI markers are shown in orange, PrE markers in red, and TE markers in blue. Black arrows indicate activation, T-shaped arrows indicate the inhibition of gene expression, and the arrow with the red cross represents unaffected expression. (Created in Biorender.com by Yan Jiao).

**Table 1 jdb-13-00033-t001:** Effects of small-molecule modulators on preimplantation development of real human embryos.

Small Molecule	Target Pathway	A./I.	Embryo Number	Treatment Duration	Concentration	Blastocyst Development Rate (Control)	ICM Marker	TE Marker	PrE Maeker	Ref.
CRT0276121	Hippo	A.	12	pre-compaction—blastocyst stage	1.5 μM	25% (83%)	→	↓	-	[29]
TRULI	Hippo	I.	5	pre-compaction—blastocyst stage	2.5 μM	100% (100%)	↑	↓	-	[30]
1-Azakenpaullone	Wnt/β-catenin	A.	68	D 3–D 5/6	20 μM	70% (86%)	→	↓	-	[31]
Wnt3	Wnt/β-catenin	A.	25	D 3–5/6	100 ng/mL	80% (87%)	→	→	-	
Cardamonin	Wnt/β-catenin	I.	77	D 3–5/6	20 μM	46% (75%)	→	↓	-	
PD0325901	FGF	I.	3	D 3–6/7	1.0 μM	-	→	-	→	[32]
PD0325901 +PD173074	FGF	I.	3	D 3–6/7	0.5 μM/100 nM	-	→	-	→	
PD173074	FGF	I.	11	D 5–D6/7	0.5 μM	-	↑	-	↓	[33]
FGF2	FGF	A.	4	D 5–D6/7	250 ng/mL	-	↓	-	↑	
SB431542	TGF-β/ACTIVIN/Nodal	I.	64	D 3–D 6	10 μM	25% (28%)	↑	-	→	[34]
Activin A	TGF-β/ACTIVIN/Nodal	A.	44	D 3–D 6	50 ng/mL	27% (28%)	→	-	→	
A8301	TGF-β/ACTIVIN/Nodal	I.	7	D 6–D 7	100 μM	-	→	-	→	[35]
BMP4	BMP	A.	-	D 3–D 6	100 ng/mL	17.4% (61.5)	→	→	→	[36]

A./I.: Activation/inhibition; -: not described; →: non-significance; ↑: significantly increased; ↓: significantly decreased.

## Data Availability

No new data were created or analyzed in this study.

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
