# Peer review of "Signaling Pathways in Human Blastocyst Development: From Molecular Mechanisms to In Vitro Optimization"

_jdb, 2025, doi:10.3390/jdb13030033_

Round 1
Reviewer 1 Report
Comments and Suggestions for Authors
The authors focused on the roles of the Hippo, Wnt/β-catenin, FGF, and TGF-β signaling pathways in the differentiation of totipotent cells in the early human embryo into trophectoderm (TE), primitive endoderm (PrE), and epiblast (EPI) during pre-implantation development. The review is comprehensive and of interest to the readers.
Major points:
- This manuscript lacks reference and discussion of the recent review on the pre-implantation embryo development: Karami et al., Signaling pathway regulators in preimplantation embryos, 2025.
- Both figures include titles but lack detailed legends. Adding comprehensive figure legends, including the used abbreviations, would enhance readability and allow readers to understand the figures without needing to refer back to the main text.
- Figure 2 is somewhat confusing as it shows separately 3 cells with cleavage cells and leaves impression that Hippo, Wnt and TGFbeta signalling occurs in different cells.
- Table is overly dense. The authors need to adopt a clearer layout to ensure the data is accessible and understandable.
Reviewer 2 Report
Comments and Suggestions for Authors
The manuscript by Jiao et al is a review article, aiming to provide a summary on the molecular mechanisms by which core signaling pathways regulate human preimplantation embryonic development. The concept is very interesting and attractive as there have been many new developments in recent years in this area. Unfortunately, the review is poorly written and difficult to read. In general, following areas require significant improvement. Firstly, since this is a review paper, please try to summarize original research findings in the paper rather than citing many previous reviews. Secondly, for a review paper, it is important to organize the raw materials (i.e. original results from research papers) in a way to support your opinions, so that the readers can easily follow. It suppose not to display all the research results one-by-one, or to obtain the conclusion from another review.
Specific points:
- Please provide a reference for the statement of ‘…8-12% couples of reproductive age….’.(line 29-30)
- In the paragraph of ZGA, it states that ZGA ‘is followed by a series of symmetrical cleavages, producing 2-cell (d.p.f 1), 4-cell (d.p.f 2), and 8-cell (d.p.f 3) embryos’ (line 74-75). Are you talking about human embryos or mouse embryos? In human ZGA mainly occur at 8-cell stage.
- Table 1 is very confusing. I am not sure what information the authors want readers to get.
- Hippo pathway:
- The title should be more informative (for example, Hippo pathway regulates lineage specification of ICM and TE), then authors should organize the contents based on this, telling a logic story rather than citing each paper sequentially.
- It would be better for the readers if the authors start this section with a description of how hippo pathway functions.
- What are the differences between human and mouse on the Hippo pathway? Do you mean that YAP/TAZ mainly partner with TEAD4 in mouse but with TEAD1 in human?
- Wnt pathway:
- Similarly, it needs a more informative title to illustrate the function of Wnt in preimplantation development.
- The whole section is written confusingly with several errors. Firstly, how could Wnt facilitate EPI from naïve to primed transition in preimplantation embryos (line 163-164)? Primed epiblast usually refer to post-implantation epiblast. The original article is wrongly stated this. Thus, authors should carefully analyze the original materials. Secondly, beta-catenin only show nuclear localization at cleavage stage under drug treatment, not naturally (line 174-175). There is no clear evidence that WNT signaling is active in preimplantation human embryos. Thirdly, reference 58 is on mice not human, thus should be not cited there (line 175-180).
Author Response
Response to the Reviewer 2’s Comments
Manuscript number: jdb-3690102
Title: Signaling Pathways in Human Blastocyst Development: From Molecular Mechanisms to In Vitro Optimization
Dear Editors and Reviewers:
We appreciate the opportunity to revise our manuscript titled “Signaling Pathways in Human Blastocyst Development: From Molecular Mechanisms to In Vitro Optimization” and are grateful for the insightful comments provided by the reviewers. These comments are all valuable and very helpful for revising and improving our paper. In the following, we have provided detailed responses to each of the reviewers’ comments. Revised portion are marked in red in the paper. Additionally, we have conducted a comprehensive revision of the entire manuscript. In this response letter, the reviewers’ comments are presented in italics, and our corresponding changes and additions to the manuscript are highlighted in red text. We have tried our best to make all the revisions clear, and we hope that the revised manuscript meets the requirements for publication.
Reviewer #2:
Comment 1: Firstly, since this is a review paper, please try to summarize original research findings in the paper rather than citing many previous reviews. Secondly, for a review paper, it is important to organize the raw materials (i.e. original results from research papers) in a way to support your opinions, so that the readers can easily follow. It suppose not to display all the research results one-by-one, or to obtain the conclusion from another review.
Response 1: Thank you for your suggestion. We have revised the manuscript by reducing the number of secondary references and instead incorporated more original research articles to support our discussion. For example, see page 5, line 119–121 and 131–132 in the revised manuscript. All revisions have been highlighted in red.
Comment 2: Please provide a reference for the statement of ‘…8-12% couples of reproductive age….’.(line 29-30)
Response 2: Thank you for noticing. We have now added the appropriate reference (Wasilewski et al., 2020) to support this statement, (page 1, line 34)
Comment 3: In the paragraph of ZGA, it states that ZGA ‘is followed by a series of symmetrical cleavages, producing 2-cell (d.p.f 1), 4-cell (d.p.f 2), and 8-cell (d.p.f 3) embryos’ (line 74-75). Are you talking about human embryos or mouse embryos? In human ZGA mainly occur at 8-cell stage.
Response 3: We apologize for the confusion. The sentence has been revised to specify the developmental timing of ZGA in human embryos. (page 2, line 76-80)
Comment 4: Table 1 is very confusing. I am not sure what information the authors want readers to get.
Response 4: As suggested, we have restructured Table 1 and added a more informative title and footnotes to improve its clarity and accessibility. (Page 4, Table 1)
Comment 5: Hippo pathway: The title should be more informative (for example, Hippo pathway regulates lineage specification of ICM and TE), then authors should organize the contents based on this, telling a logic story rather than citing each paper sequentially.
Response 5: We have revised the Hippo signaling section by updating the title to better reflect its function, and have done the same for the WNT, FGF, and TGF-β sections. (Page 4, line 108; Page 6, line 168; Page 6, line 208; Page 7, line 261)
Comment 6: It would be better for the readers if the authors start this section with a description of how hippo pathway functions.
Response 6: We have revised the Hippo signaling section by start this section with a description of how it functions, and have done the same for the WNT, FGF, and TGF-β sections. (Page 4-5, line 109-118; Page 6, line 169-178; Page 6, line 209-217; Page 7, line 265-266)
Comment 7: What are the differences between human and mouse on the Hippo pathway? Do you mean that YAP/TAZ mainly partner with TEAD4 in mouse but with TEAD1 in human?
Response 7: We apologize for the confusion in our original text. We have now clarified the species-specific differences in the Hippo signaling pathway during early lineage segregation. Specifically, TEAD4 plays a crucial role in the first lineage segregation in mice but may function differently in humans. In human embryos, TEAD1 and YAP1 are co-localized in TE and PrE precursor cells, suggesting a possible role in the second lineage segregation. The relevant section has been fully revised to reflect these distinctions more accurately and is now clearly presented in the revised manuscript (page 5, lines 131-155).
Comment 8: Wnt pathway: Similarly, it needs a more informative title to illustrate the function of Wnt in preimplantation development. The whole section is written confusingly with several errors. Firstly, how could Wnt facilitate EPI from naïve to primed transition in preimplantation embryos (line 163-164)? Primed epiblast usually refer to post-implantation epiblast. The original article is wrongly stated this. Thus, authors should carefully analyze the original materials.
Response 8: Thank you for your valuable comment. We sincerely apologize for the misinterpretation. In the revised manuscript, we have corrected this error and revised the corresponding section to ensure an accurate representation. (page 6, lines 181-183).
Comment 9: Secondly, beta-catenin only show nuclear localization at cleavage stage under drug treatment, not naturally (line 174-175). There is no clear evidence that WNT signaling is active in preimplantation human embryos.
Response 9: Thank you for your careful review and valuable suggestions. We sincerely apologize for our misinterpretation of the original studies. We have corrected these inaccuracies in the revised manuscript (page 6, lines 200-201).
Comment 10: Thirdly, reference 58 is on mice not human, thus should be not cited there (line 175-180).
Response 10: Thank you for your valuable comment. Although the main research subjects of reference 58 are mice, the latter part of the paper includes experimental validation using human embryos, supporting the conclusion that inhibition of WNT signaling during the peri-implantation period promotes embryo development. Therefore, we choose to citing this reference here.
We tried our best to improve the manuscript and made some changes marked in red in revised paper which will not influence the content and framework of the paper. We appreciate for editors and reviewer’s warm work earnestly, and hope the correction will meet with approval. Best regards!
Sincerely,
Lifeng Xiang

Reviewer 3 Report
Comments and Suggestions for Authors
Review
In this review article, the authors aim to systematically analyze the molecular mechanisms involved in blastocyst lineage differentiation and morphogenesis, and to assess the feasibility of optimizing in vitro culture by targeting key signaling nodes. However, several points need to be addressed to improve the clarity and overall impact of the manuscript.
Major comments
- In several sections, the authors do not sufficiently distinguish between species and often mix references from different organisms. Since preimplantation development varies significantly among species—for example, in the timing of zygotic genome activation (ZGA) and compaction—it is essential to clearly specify the species being discussed, especially when referring to non-human studies. For instance, the roles of the Hippo signaling pathway are described in the context of mouse embryos (line 98, reference 28; line 114, references 35–38; line 123, references 39–41) without explicitly indicating that these findings are from mice.
- Although the authors state that they aim to assess the feasibility of optimizing in vitro culture by targeting key signaling nodes, they do not provide concrete examples or specific strategies for how in vitro culture conditions could be improved for human embryos.
Minor comments
- Line 40: Please replace the colon “:” with a period “.”
- Line 53: Please remove the space and period before “[11–13]”.
- Lines 63–64: The phrase “So as to provide a theoretical basis for improving the in vitro culture system and enhancing clinical outcomes in ART” is not a complete sentence grammatically. Please revise it or incorporate it into the preceding sentence.
- Line 72: To avoid confusion, please standardize the terms “embryonic genome activation” and “zygotic genome activation” (used in line 48), or clarify that they are used interchangeably.
- Line 77: Please insert a space between “compaction” and the citation “[20]”.
- Line 106: Please insert a space between “formation” and “(Table 1)”.
- Page 4: Please provide a descriptive title for Table 1.
- Table 1: The “Reference” heading in the top row should be removed and replaced with reference numbers placed in the appropriate cells or as footnotes.
- Lines 279–289: Please provide appropriate references to support the statements in this section.
Author Response
Response to the Reviewer 3’s Comments
Manuscript number: jdb-3690102
Title: Signaling Pathways in Human Blastocyst Development: From Molecular Mechanisms to In Vitro Optimization
Dear Editors and Reviewers:
We appreciate the opportunity to revise our manuscript titled “Signaling Pathways in Human Blastocyst Development: From Molecular Mechanisms to In Vitro Optimization” and are grateful for the insightful comments provided by the reviewers. These comments are all valuable and very helpful for revising and improving our paper. In the following, we have provided detailed responses to each of the reviewers’ comments. Revised portion are marked in red in the paper. Additionally, we have conducted a comprehensive revision of the entire manuscript. In this response letter, the reviewers’ comments are presented in italics, and our corresponding changes and additions to the manuscript are highlighted in red text. We have tried our best to make all the revisions clear, and we hope that the revised manuscript meets the requirements for publication.
Reviewer #3:
Comment 1: In several sections, the authors do not sufficiently distinguish between species and often mix references from different organisms. Since preimplantation development varies significantly among species—for example, in the timingof zygotic genome activation (ZGA) and compaction—it is essential to clearly specify the species being discussed, especially when referring to non-human studies. For instance, the roles of the Hippo signaling pathway are described in the context of mouse embryos (line 98, reference 28; line 114, references 35–38; line 123, references 39–41) without explicitly indicating that these findings are from mice.
Response 1: Thank you for your thoughtful comment. We apologize for the oversight and the lack of clarity in distinguishing between species. We now realize that some of the referenced studies were not clearly identified as being based on non-human models, particularly mouse embryos. We have carefully revised the relevant sections throughout the manuscript to explicitly indicate the species being discussed and to avoid confusion between human and non-human data, as shown on page 3, line 99-103 and page 5, line 119-130 of the revised manuscript.
Comment 2: Although the authors state that they aim to assess the feasibility of optimizing in vitro culture by targeting key signaling nodes, they do not provide concrete examples or specific strategies for how in vitro culture conditions could be improved for human embryos.
Response 2: Thank you for your helpful comment. We have now added specific examples and potential strategies for optimizing in vitro culture conditions by targeting key signaling pathways in human embryos, as suggested (page 5, line 164-167; page 8, line 290-293).
Comment 3: Line 40: Please replace the colon “:” with a period “.”
Response 3: We were very sorry for our careless mistakes. Thank you for your reminder (page 1, lines 44).
Comment 4: Line 53: Please remove the space and period before “[11–13]”.
Response 4: Thanks for your correction, it has been revised (page 2, line 57).
Comment 5: Lines 63–64: The phrase “So as to provide a theoretical basis for improving the in vitro culture system and enhancing clinical outcomes in ART” is not a complete sentence grammatically. Please revise it or incorporate it into the preceding sentence.
Response 5: Thank you for your careful review. We have revised the sentence by incorporating the phrase into the preceding sentence to ensure grammatical correctness and improve clarity (page 3, line 67-69).
Comment 6: Line 72: To avoid confusion, please standardize the terms “embryonic genome activation” and “zygotic genome activation” (used in line 48), or clarify that they are used interchangeably.
Response 6: Thank you for your valuable suggestion. To avoid confusion, we have standardized the terminology throughout the manuscript by using “zygotic genome activation (ZGA)” consistently instead of “embryonic genome activation (page 2, line 78).
Comment 7: Line 77: Please insert a space between “compaction” and the citation “[20]”.
Response 7: We were sorry for our careless mistakes. Thank you for your reminder. The space has been inserted (page 2, lines 80-81).
Comment 8: Line 106: Please insert a space between “formation” and “(Table 1)”.
Response 8: We sincerely thank you for your careful reading. As suggested, we have corrected it (page 3, lines 103).
Comment 9: Page 4: Please provide a descriptive title for Table 1.
Response 9: Thank you for your comment. We have added a descriptive title to Table 1 to clarify its content (page 4, Table 1).
Comment 10: Table 1: The “Reference” heading in the top row should be removed and replaced with reference numbers placed in the appropriate cells or as footnotes.
Response 10: Thank you for your helpful suggestion. We have removed the “Reference” heading from the top row of Table 1 and have placed the reference numbers directly in the appropriate cells instead (page 4, Table 1).
Comments11: Lines 279–289: Please provide appropriate references to support the statements in this section.
Response 11: Thank you for your suggestion. We have now added appropriate references to support the statements in page 8, line 312 and page 8, line 315.
We tried our best to improve the manuscript and made some changes marked in red in revised paper which will not influence the content and framework of the paper. We appreciate for editors and reviewer’s warm work earnestly, and hope the correction will meet with approval. Best regards!
Sincerely,
Lifeng Xiang

Round 2
Reviewer 1 Report
Comments and Suggestions for Authors
The authors had addressed in full my comments on the manuscript.
Author Response
Response to the Editor and Reviewers
Manuscript number: jdb-3690102
Title: Signaling Pathways in Human Blastocyst Development: From Molecular Mechanisms to In Vitro Optimization
Dear Editors and Reviewers:
We would like to express our sincere gratitude to the editor and reviewers for their time, effort, and valuable comments on our manuscript. And we have carefully addressed all the suggestions provided. Our detailed responses are as follows:
Reviewer #1
Comment: “The authors had addressed in full my comments on the manuscript.”
Response: We are truly grateful for your positive feedback and recognition of our revision.
Reviewer 2 Report
Comments and Suggestions for Authors
The authors have addressed all the concerns raised by the reviewers. There is no further comments. One minor suggestion is to place the title of the table above the table rather than below the table, which is different from figure legends.
Author Response
Response to the Editor and Reviewers
Manuscript number: jdb-3690102
Title: Signaling Pathways in Human Blastocyst Development: From Molecular Mechanisms to In Vitro Optimization
Reviewer #2
Comment: “The authors have addressed all the concerns raised by the reviewers. There is no further comments.”
Response: We sincerely appreciate your encouraging remarks and confirmation that our revision has addressed the concerns.
Comment: “One minor suggestion is to place the title of the table above the table rather than below the table, which is different from figure legends.”
Response: Thank you for this helpful suggestion. We have revised the manuscript accordingly by moving the table titles above the corresponding tables. (page 4, Table 1).

Reviewer 3 Report
Comments and Suggestions for Authors
Thank you for your revison.
Author Response
Response to the Editor and Reviewers
Manuscript number: jdb-3690102
Title: Signaling Pathways in Human Blastocyst Development: From Molecular Mechanisms to In Vitro Optimization
Dear Editors and Reviewers:
We would like to express our sincere gratitude to the editor and reviewers for their time, effort, and valuable comments on our manuscript. And we have carefully addressed all the suggestions provided. Our detailed responses are as follows:
Reviewer #3
Comment: “Thank you for your revision.”
Response: We sincerely thank you for your kind feedback.